# Complex Event Schema Induction with Knowledge-Enriched Diffusion Model

**Yupu Hao[1,2], Pengfei Cao[1,2], Yubo Chen[1,2], Kang Liu[1,2], Jiexin Xu[3],**
**Huaijun Li[3], Xiaojian Jiang[3], Jun Zhao[1,2]**

[1] The Laboratory of Cognition and Decision Intelligence for Complex Systems,
Institute of Automation, Chinese Academy of Sciences, Beijing, China
[2] School of Artificial Intelligence, University of Chinese Academy of Sciences, Beijing, China
[3] China Merchants Bank
haoyupu2023@ia.ac.cn, {pengfei.cao, yubo.chen, kliu, jzhao}@nlpr.ia.ac.cn

## Abstract

The concept of a complex event schema pertains to the graph structure that represents real-world knowledge of events and their multi-dimensional relationships. However, previous studies on event schema induction have been hindered by challenges such as error propagation and data quality issues. To tackle these challenges, we propose a knowledge-enriched discrete diffusion model. Specifically, we distill the abundant event scenario knowledge of Large Language Models (LLMs) through an object-oriented Python style prompt. We incorporate this knowledge into the training data, enhancing its quality. Subsequently, we employ a discrete diffusion process to generate all nodes and links simultaneously in a non-auto-regressive manner to tackle the problem of error propagation. Additionally, we devise an entity relationship prediction module to complete entity relationships between event arguments. Experimental results demonstrate that our approach achieves outstanding performance across a range of evaluation metrics.[1]

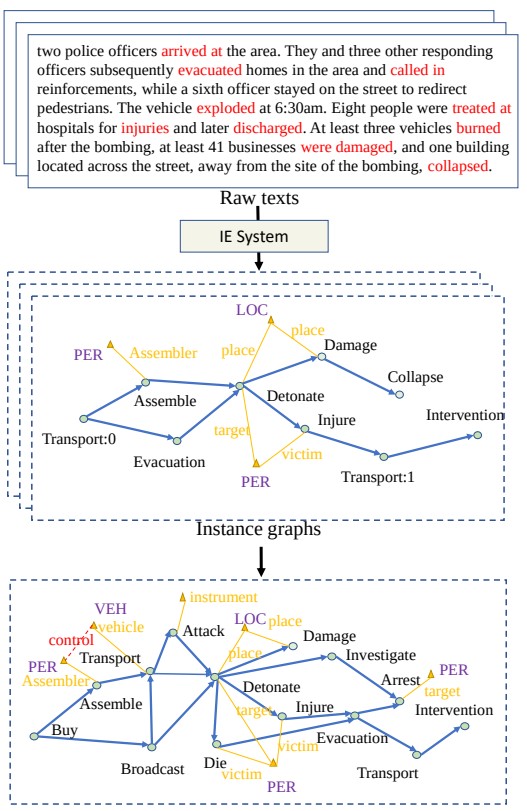

Figure 1: An example of schema induction process for complex event "*Bombing*".

## 1 Introduction

Event schema induction aims to summarize common patterns and structures from historical events. Current studies mainly induce the atomic schema for each independent event type and their arguments separately (e.g. "*Attack*" event with the arguments: "*Attacker*", "*Target*", "*Instrument*" and "*Place*"), without considering the correlation between events (Chambers and Jurafsky, 2008; Chambers, 2013; Nguyen et al., 2015). However, some real-world events are usually very complex, consisting of multiple events and their relations. For example in Figure 1, *Bombing* is a complex event, which involves some fine-grained events, such as *Assemble*, *Detonate* and *Injure*. Therefore, some

researchers attempt to study the *complex event schema induction* task, which abstracts typical structures for complex events from event data. Figure 1 illustrates an example of the complex event schema induction process for the scenario of *Bombing*. Initially, an information extraction (IE) tool (Du et al., 2022) is utilized to extract instance graphs from raw texts. Subsequently, we induce the event schema based on these extracted instance graphs. The resulting event schema is represented as a graph, where events are interconnected through temporal links (e.g., *Damage* occurs after *Detonate*) and their argument relations (e.g., the *target*

---

[1]Code is available at https://github.com/hypasd-art/KDM/

of the *Detonate* event assumes the *victim* role in the subsequent *Injure* event).

However, inducing complex event schema is non-trivial. As shown in Figure 1, it necessitates the model's ability to summarize the events within instance graphs and possess a profound understanding of the multi-dimensional relationships between these events. Recently, graph-based methods are proposed for this task by utilizing graph generation techniques (Li et al., 2021; Jin et al., 2022). For example, Li et al. (2021) proposes an auto-regressive generation method that generates the schema following event temporal order. Similarly, Jin et al. (2022) leverages an auto-encoder to encode the global skeleton information and decode the schema graph event by event. Despite successful efforts, these methods still face two critical challenges:

**Knowledge Coverage of Instance Graphs**: The event schema induction task summarizes the instance graphs to obtain the event schema. Thus, the quality of the instance graphs is crucial for the event schema induction. However, the instance graphs are extracted via Information Extraction (IE) tools (Rui et al., 2022), whose knowledge coverage is very limited. For example, as the representative IE tool, RESIN (Wen et al., 2021) is trained on fixed datasets and can only extract predefined types of entities and events. Besides, the extraction performance of RESIN is also unsatisfactory, which only achieves approximately 64% of F1-score for event detection on the ACE dataset. It indicates that the IE tool is difficult to extract complete instance information, even for predefined event types. Therefore, how to improve the knowledge coverage of instance graphs is an important problem.

**Error Propagation of Auto-regressive Decoding**: Previous graph-based approaches are based on the auto-regressive generation manner (Li et al., 2021; Jin et al., 2022), generating the entire event schema graph node by node, which may lead to error accumulation over time and therefore degrade the generation performance. For example, in Figure 1, the model may mistakenly generate "Injure" instead of "Detonate" leading to the omission of subsequent events such as "Damage" and "Investigate" in the generated schema or resulting in incorrect nodes being generated in the next. The final generated event schema graph will consist of dozens of nodes and edges at the minimum, as each instance graph used for training contains an average of 117 event nodes and 246 temporary links according to our statistics on the Suicide-IED dataset (Li et al., 2021). The need to generate so many nodes and edges will inevitably exacerbate the problem of error accumulation. Thus, it is essential to address the error propagation problem during schema graph generation.

In this paper, we propose a novel method termed as **K**nowledge-Enriched **D**iffusion **M**odel (**KDM**) to address aforementioned problems. Firstly, to improve the knowledge coverage of instance graphs, we devise a *Instance Graph Expansion* module. As Large Language Models (LLMs) are trained on vast corpora of texts (Touvron et al., 2023; Zhao et al., 2023; Wang et al., 2023) and therefore possess extensive event and entity knowledge of the real world, we leverage the LLMs (Chowdhery et al., 2022; Ouyang et al., 2022) as the knowledge databases to inject knowledge into instance graphs. The module utilizes a Python style object-oriented prompt to extract event knowledge from LLMs, and adds the knowledge into the instance graphs. Secondly, to tackle error propagation of auto-regressive decoding, we propose an *Event Skeleton Generation* module, which utilizes discrete diffusion model to predict all nodes and links simultaneously in non-auto-regressive manner but not generates individually based on time series, which alleviate the error propagation problem (Austin et al., 2021; Yang et al., 2023; Vignac et al., 2022). Finally, we devise an *Entity Relation Prediction* module, which expands the event skeleton with corresponding arguments and predicts their relations to get a complete schema.

The contributions of our work include: (1) We propose a **K**nowledge-Enriched discrete **D**iffusion **M**odel (**KDM**) for complex event schema induction task. To the best of our knowledge, we are the first to simultaneously utilize LLMs and diffusion models to accomplish the task. (2) To improve knowledge coverage of instance graphs, we propose an Instance Graph Expansion module, which distillates the event knowledge in LLMs with python code-style prompt. To solve the error propagation problem, we design an event skeleton generation module, which predicts all nodes and links simultaneously. (3) We conduct extensive experiments on three widely used datasets. Experimental result indicates that our proposed method outperforms state-of-the-art baselines.

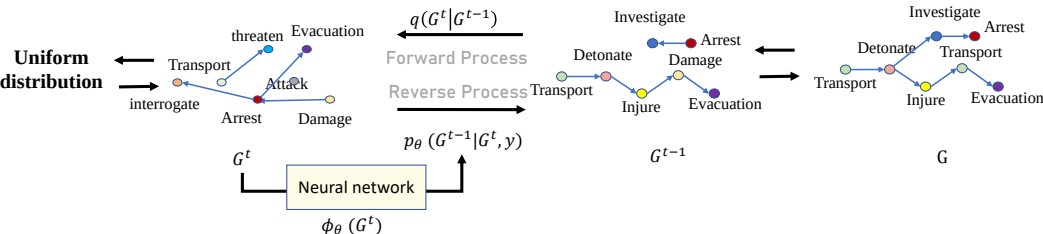

Figure 2: The discrete diffusion process. In forward process, the noise changes the types of nodes and edges.

## 2 Preliminaries and Problem Formulation

### 2.1 Preliminaries

Discrete diffusion model preserves the discrete characteristics of each element in the training data $x_0$, it introduces noise to each element $x_0^r \in x_0$ to into the uniform distribution and reverses them by removing the noise (Austin et al., 2021). Figure 2 shows the process of graph-based discrete diffusion.

**The forward process.** This process progressively adds noise to $x_0$ by transition probability matrix $Q_t$ at $t$ step.

$$q(x_t|x_{t-1}) = x_{t-1}Q_t \quad (1)$$

where $|Q_t|_{ij} = q(x_t = j|x_{t-1} = i)$ indicates the probability of transition from $x_{t-1} = i$ to $x_t = j$. The forward process gradually converts each $x_0^r \in x_0$ to a uniform distribution when $T$ is large enough.

**The reverse process.** Reverse process $p_\theta$ with learnable parameters $\theta$ aims to convert the noise distribution $x_T$ back to the original $x_0$:

$$p_\theta(x_{0:T}) = p_\theta(x_T) \prod_{t=1}^{T} p_\theta(x_{t-1}|x_t) \quad (2)$$

$$p_\theta(x_{t-1}|x_t) = \int q(x_{t-1}|x_t, x_0)dp_\theta(x_0|x_t) \quad (3)$$

and according to Bayes formula as follows:

$$q(x_{t-1}|x_t, x_0) = \frac{q(x_t|x_{t-1}, x_0)q(x_{t-1}|x_0)}{q(x_t|x_0)} \quad (4)$$

Therefore, the task becomes predicting $p_\theta(x_0|x_t)$ using a neural network.

### 2.2 Problem Formulation

In the instance graphs about a specific topic $y$ (e.g., *Bombing*), nodes represent events and entities, while edges have three types: the temporal link, the argument link and the entity relation link. The instance graph is denoted as $\mathcal{G} = (\mathcal{N}, \mathcal{E})$, We define the distribution of graph as $G = (N, E)$, where node set $\mathcal{N}$ is sampled from node feature distribution $N \in \mathcal{R}^{n \times a}$ and edge set $\mathcal{E}$ is sampled from edge feature distribution $E \in \mathcal{R}^{n \times n \times b}$. Here, $c(p)$ represents the one-hot vector (category scalar) sampled based on probability distribution $p$. At time $t$, the instance graph is defined as $\mathcal{G}^t = (\mathcal{N}^t, \mathcal{E}^t) = (c(N^t), c(E^t))$.

The objective of this task is to learn an event schema $\mathcal{S}_y$ from a set of instance graphs $\mathcal{D}_y = \{\mathcal{G}(1), \mathcal{G}(2), \ldots, \mathcal{G}(m)\}$ that belong to the same specific topic.

## 3 Our Approach

To solve the complex event schema induction task, we propose a Knowledge-Enriched Discrete Diffusion Model, as shown in Figure 3. Our method mainly consists of three modules: (1) Instance Graph Expansion, which expands the instance graphs using the complex event knowledge obtained from LLMs atomically. (2) Event Skeleton Induction, which summarizes the event evolution skeleton using a discrete diffusion model. (3) Entity Relation Prediction, which decorates the arguments to the event skeleton and then use a simple graph transformer to predict the entity relations. We will illustrate each component in detail.

### 3.1 Instance Graph Expansion

In this section, we will illustrate how to obtain knowledge about event schemas from LLMs and inject them into instance graphs. Complex event schemas involve intricate graph structures, while LLMs are good at processing unstructured language tasks. To retain structured information of instance graphs, we need LLMs to be able to handle structured inputs and outputs. Considering the powerful coding capabilities of LLMs, we treat events as Python objects. In detail, events, entities, and their intricate relations can correspond to classes, attributes, and instances in the object-

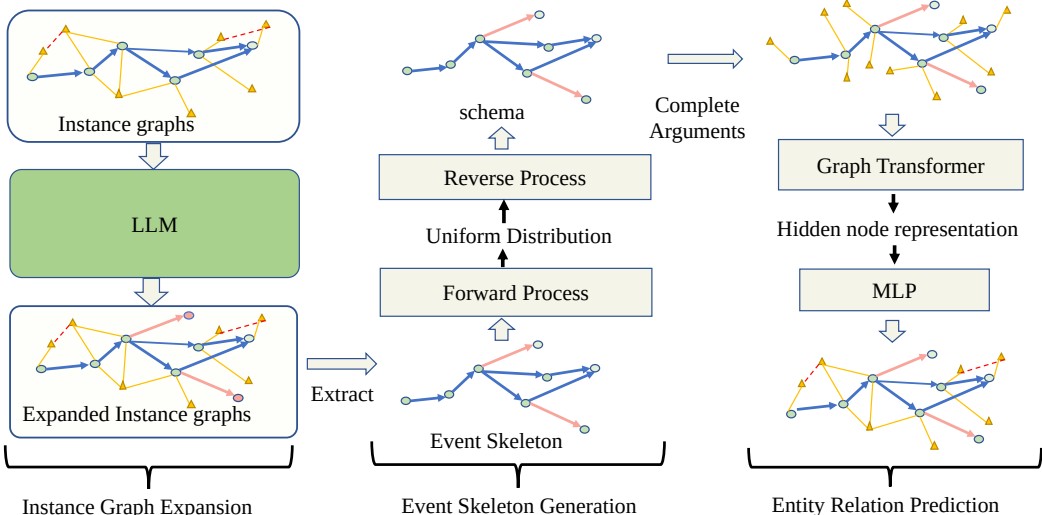

Figure 3: The model structure. The data passes through Instance Graph Expansion module, Event Skeleton Generation module, and Entity Relation Prediction module in sequence to obtain the final schema.

oriented paradigm, respectively. This module includes three aspects: event knowledge expansion, temporal relation expansion, and entity relation expansion.

In event knowledge expansion, we select frequently occurring event sequences from the training instance graphs and write them into Python classes. Then we ask the LLM to enrich Python code. In this way, we will obtain new classes which represent new events that have a high correlation with the scenario. We filter out new events that occur less frequently than a hyperparameter $K$ and are not in the predefined event categories. In temporal relation expansion, we write the obtained events as multiple-choice questions to establish their temporal relation with existing event sequences. In entity relation expansion, we obtain the argument connection relation between the new and existing events by encoding new events into Python code and instantiating the class. By effectively leveraging the complex event knowledge contained in LLMs, our approach enhances the event schema generation process. The details and examples refer to Appendix E.

## 3.2 Event Skeleton Generation

In this section, we will introduce the forward and reverse processes of discrete diffusion based on the instance graphs through Instance Graph Expansion module. We have adopted the diffusion framework of Vignac et al.(2022) and made improvements based on it. Here, we denote the distribution $G$ at

time $t$ as $G_t$.

**The forward diffusion process**. In this process, we apply noise separately on each node and edge. This is achieved by multiplying the node and edge distributions with the transition probability matrix $Q$. By doing so, we can obtain the graph $\mathcal{G}_t$ from the previous graph $\mathcal{G}_{t-1}$. Mathematically, this can be expressed as:

$$q(G_t|G_{t-1}) = (N_{t-1}Q_t^N, E_{t-1}Q_t^E)$$
$$= (N\bar{Q}_t^N, E\bar{Q}_t^E) \quad (5)$$

Where $\bar{Q}_t = Q_0Q_1\cdots Q_t$. The transition probability matrix $Q_t$ is defined as $\alpha^t I + (1 - \alpha^t)\mathbf{1}(\mathbf{1})^T/K$, where $\mathbf{1}$ is a column vector of all ones, and $\alpha^t$ varies from 1 to 0 (Austin et al., 2021). This formulation ensures that the distribution $q(G_t|G_0)$ consistent with uniform distribution when time $t$ becomes sufficiently large.

Next, we sample the node and edge types from these probability distributions to obtain a discrete graph:

$$\mathcal{G}_t = c(q(G_t|G_{t-1})) \quad (6)$$

**The reverse diffusion process**. We aim to remove the noise from the graphs using a parameterized reverse process $p_\theta$. Following the formulation presented by Austin et al.(2021), we can express the posterior $p_\theta(G_{t-1}|G_t)$ as:

$$p_\theta(G_{t-1}|G_t) = \int q(G_{t-1}|G_t, G_0)dp_\theta(G_0|G_t) \quad (7)$$

To predict the clean graph distribution $G_p^t = p_\theta(G_0|G_t)$ at time $t$ given the noisy input $\mathcal{G}_t$, we

train a graph transformer $\phi_\theta$ that outputs the clean graph representation:

$$G_p^t = (N_p^t, E_p^t) = \phi_\theta(\mathcal{G}_t) \tag{8}$$

Our model $\phi_\theta$ adopt transformer structure (Vaswani et al., 2017). Previous graph transformer model (Ying et al., 2021) is not appropriate for encoding directed graphs based on time series, because the relative position information between nodes is lost during the noise adding process. For instance, the self-attention mechanism module in the transformer cannot differentiate two "transport" events that occur in different time periods. To address this issue, we encode the depth information of event nodes as a fixed-feature embedding $n_{dep}$ into the model. Before inputting the graph into the transformer, we add the depth feature to the corresponding node feature.

The depth fixed-feature embedding is encoded as follows:

$$n_{dep} = \begin{cases} \sin(w_k \cdot n_d), & i = 2k \\ \cos(w_k \cdot n_d), & i = 2k+1 \end{cases} \tag{9}$$

where $w_k = 1/10000^{2k/n_d}$ and $n_d$ is the average depth of node $n$, $i$ is the index of the depth embedding.

Inspired by Vignac et al.(2022), our transformer model comprises several layers, each of which consists of a self-attention module and a feed-forward network. For layer $l$, the self-attention module takes as input time features $t$, node features $N_l^t$, edge features $E_l^t$, and updates their representation as follows:

$$Att_l = E_l^t W_a + E_l^t W_m \odot N_l^t W_Q (N_l^t W_K)^T \tag{10}$$

$$N_{l+1}^t = MLP(t W_{tn} + Att_l N_l^t W_V) \tag{11}$$

$$E_{l+1}^t = MLP(t W_{te} + Att_l) \tag{12}$$

where $\odot$ denotes the pairwise multiplication. $W_a$, $W_m$, $W_{tn}$, $W_{te}$, $W_Q$, $W_K$, $W_V$ are trainable parameters.

To optimize our model, we use the cross-entropy loss $\mathcal{L}_{CE}$ weighted by $\lambda$:

$$\mathcal{L}_{CE}(G_p^t, G) = \sum_{\mathcal{G} \in \mathcal{D}} CE(N_p^t, N) + \lambda CE(E_p^t, E) \tag{13}$$

Once we obtain the clean graph distribution $G_p^t$, we can infer the node distribution $p_\theta(n_{t-1}|n_t)$ and edge distribution $p_\theta(e_{t-1}|e_t)$ using the equations:

$$p_\theta(n_{t-1}|n_t) = \sum_{n_0=1}^{K_n} q(n_{t-1}|n_t, n_0) p_\theta(n_0|n_t)$$

$$p_\theta(e_{t-1}|e_t) = \sum_{e_0=1}^{K_e} q(e_{t-1}|e_t, e_0) p_\theta(e_0|e_t) \tag{14}$$

where $K_n$ is the node type number, and $K_e$ is edge type number. Before the next reverse process, we will get the discrete graph $\mathcal{G}_{t-1}$ from its distribution by probability sampling.

Our model obtains the final event schema $\mathcal{G}$ through T-step reversing process in non-AR manner. For further algorithm and derivation details, please refer to Appendix C.

**Conditional Generation**. Previous approaches (Li et al., 2020, 2021; Jin et al., 2022) need to train separate models for each scenario to ensure accurate generation. However, in order to generate event schemas for various scenarios using a single model and improve the model's generalization capabilities, we also propose the conditional diffusion model named as **KDMall** as a supplement.

We incorporate the category information $y$ of the instance graphs as an additional attribute to control the training process of the model (Ho and Salimans, 2022; Dhariwal and Nichol, 2021). This allows us to influence the category of the generated schema. The formulation is as follows:

$$p_\theta(G_{t-1}|G_t, y) = \int q(G_{t-1}|G_t, G_0) p_\theta(G_0|G_t, y) \tag{15}$$

Therefore, we only need to encode the category information $y$ into the neural network. We simply concatenate it into temporal features, enabling the conditional diffusion model to generate event schemas of different categories:

$$G_p^t = (N_p^t, E_p^t) = \phi_\theta(\mathcal{G}_t, y) \tag{16}$$

### 3.3 Entity Relation Prediction

In this module, we have developed a simple architecture that combines a graph transformer for obtaining node representations with an MLP layer for relation prediction. This module takes the event skeleton, expanded with event argument roles, as input and generates the complete event schema by predicting the relations between entities.

While previous models have primarily focused on entity types in the classification process, neglecting the significance of events and event roles, we address this limitation by artificially aggregating them together. We initialize the node features using BERT model (Devlin et al., 2018). Specifically, for each event or entity node $n_i$, $BERT(n_i)$ represents its type embedding encoded by BERT. For entity nodes, $n_i^e$ indicates the event node that entity node $n_i$ belongs to, and $n_i^r$ is a fixed embedding representing the role played by entity node $n_i$ in

event $n_i^e$, The encoding formula of embeddings $\boldsymbol{n}_i^r$ is the same as that of equation 9.

$$\boldsymbol{n}_i = \mathrm{concat}(BERT(n_i), BERT(n_i^e)) \qquad (17)$$

Our transformer encoder is the same as the model used for Event Skeleton Generation, except that it lacks the time feature. The graph transformer outputs $\hat{\boldsymbol{n}}_i$ corresponding to the input $\boldsymbol{n}_i$, which is then passed to the MLP predictor.

The predicted relation type $\boldsymbol{r}_{ij}$ of entity node $n_i$ and $n_j$ is then computed as:

$$\boldsymbol{r}_{ij} = MLP([\hat{\boldsymbol{n}}_i + \boldsymbol{n}_i^\tau, \hat{\boldsymbol{n}}_j + \boldsymbol{n}_j^\tau]) \qquad (18)$$

It is worth noting that the classification problem is highly unbalanced. To address this issue, we set different weights for different categories of the loss function, defined as:

$$\mathcal{L} = \sum_{entity\ i,j} H(\hat{\boldsymbol{r}}_{ij})CE(\hat{\boldsymbol{r}}_{ij}, \boldsymbol{r}_{ij}) \qquad (19)$$

where $\hat{\boldsymbol{r}}_{ij}$ denotes the true relation between entity $i$ and $j$ and $H(\cdot)$ is a scalar function that assigns balanced weights to different relationships, with each relationship corresponding to a specific value.

## 4 Experiments

### 4.1 Datasets

We conduct experiments using the IED Schema Learning Corpus released by Li et al.(2021). The dataset utilizes the DARPA KAIROS ontology. The corpus specifically focuses on three sub-types of complex events related to Improvised Explosive Devices (IEDs): General-IED, Mass-Car-Bombing-IED, and Suicide-IED.

However, the test data in the corpus has data quality issues since it is also extracted through IE tools. To address this, we manually modify the test data, generating golden test event schemas based on the modified data. Additionally, to ensure the objective evaluation of our model's effectiveness, we record the test results using the original, unmodified data, which are provided in Appendix 7.

### 4.2 Baselines

In this work, we compare the proposed event schema induction model with two baselines:

**Frequency-Based Sampling (FBS) model** which constructs the event schema according to frequency distributions of temporal links in the training data. At each timestamp, FBS samples a pair of event types according to their frequency and adds the sampled edge into the schema graph. The procedure is repeated until FBS detects a cycle in the schema graph after adding a new edge.

**Double Graph Auto-encoders Model (Double-GAE)** (Jin et al., 2022), the state-of-the-art schema induction model which designs a variational directed acyclic graph auto-encoder to extract the event skeleton. Then it uses another GCN based auto-encoder to reconstruct entity-entity relations.

**Large Language Model (LLM)** have strong understanding and generation abilities, We ask the large language model (ChatGPT) to directly generate the event schema and use it as the baseline.

### 4.3 Evaluation Metrics

To evaluate the quality of the generated schema, we compare the schema with test instance graphs in terms of the following metrics to see how well the schema match real world instance graphs, the following evaluation metrics are employed:

(1) Event type match. we calculate the F1 score between the event types present in the schema graph and test instance graphs.

(2) Event sequence match. A good schema is able to track events through a timeline. we calculate the F1 score between the event sequences of length 2 or 3 present in the schema graph and the test instance graphs.

(3) Node/edge type distribution. we compare the Kullback-Leibler (KL) divergence of the node and edge type distributions between the schema graph and each test instance graph.

(4) Event Argument Connection Match (CM). Complex event graph schema includes entities and their relations, representing how events are connected through arguments. Because there is a serious long tail issue with the data, we calculate the macro F1-score for every pair of relationships between entities.

### 4.4 Overall Results

As shown in Table 1, the result demonstrates the effectiveness of **KDM** in capturing important events and their relationships. Specifically, our approach outperforms the baseline methods in terms of event sequence matching, particularly for longer path lengths (l=3).

These improvements can be attributed to the discrete diffusion process employed in our model. This process allows our model to simultaneously predict the categories of all nodes and edges, making it well-suited for graph generation. Addition-

| Dataset | Model | Event type Match(F1) | Event sequence match | | KL divergence | | CM |
|---|---|---|---|---|---|---|---|
| | | | l=2 | l=3 | Node-type | Edge-type | |
| General-IED | FBS | 0.614 | 0.199 | 0.064 | 2.98 | 6.13 | - |
| | DoubleGAE | 0.627 | 0.266 | 0.093 | 2.43 | 5.57 | 0.046 |
| | LLM | 0.520 | 0.176 | 0.041 | 2.72 | 5.84 | - |
| | **KDM (ours)** | **0.704** | **0.380** | **0.181** | **2.32** | **4.53** | **0.185** |
| Car-IED | FBS | 0.650 | 0.198 | 0.065 | 1.86 | 5.85 | - |
| | DoubleGAE | 0.654 | 0.285 | 0.107 | 2.12 | 5.71 | 0.044 |
| | LLM | 0.515 | 0.150 | 0.031 | 2.70 | 6.34 | - |
| | **KDM (ours)** | **0.701** | **0.395** | **0.207** | **1.91** | **4.16** | **0.176** |
| Suicide-IED | FBS | 0.626 | 0.210 | 0.061 | 2.11 | 5.59 | - |
| | DoubleGAE | 0.624 | 0.272 | 0.096 | 2.19 | 5.33 | 0.046 |
| | LLM | 0.493 | 0.174 | 0.053 | 2.75 | 5.65 | - |
| | **KDM (ours)** | **0.713** | **0.462** | **0.268** | **1.91** | **3.45** | **0.176** |

Table 1: Schema matching score (%) is calculated by checking the intersection of the induced schemas and the manually checked test schemas.

| Dataset | Model | Event type Match(F1) | Event sequence match | |
|---|---|---|---|---|
| | | | l=2 | l=3 |
| General-IED | **KDM** | 0.704 | 0.380 | 0.181 |
| | **KDMall** | **0.723** | **0.413** | **0.202** |
| Car-IED | **KDM** | 0.701 | **0.395** | **0.207** |
| | **KDMall** | **0.703** | 0.389 | 0.198 |
| Suicide-IED | **KDM** | **0.714** | **0.462** | **0.268** |
| | **KDMall** | 0.714 | 0.450 | 0.251 |

Table 2: Schema matching score (%) by checking the intersection of induced schemas and manual checked test schemas. **KDMall** is the conditional diffusion model trained on three IED datasets.

ally, the Transformer architecture leveraged in our model effectively utilizes global features through the self-attention mechanism, resulting in improved prediction accuracy.

Furthermore, our model shows remarkable improvements in the Connection Match evaluation, indicating the effectiveness of our graph transformer model than GCN graph auto-encoder in Double-GAE.

## 4.5 Conditional Generation Results

Building upon the aforementioned diffusion model, in order to improve the possibility of the model's generalization ability, we present an extension in the form of a conditional diffusion model as a supplement. This model enables the generation of event schemas for various scenarios using a single model.

As shown in Table 2, when comparing **KD-Mall** with our model trained on a specific-dataset, we find that **KDMall** shows improved generalization capabilities and better understanding of event relationships, particularly in the "General-IED" scenario. Additionally, in other datasets, **KD-Mall** demonstrates comparable results to the model trained on a single dataset, indicating the potential of our conditional generation process. The incorporation of diverse training data enables the model to learn common patterns and associations across different scenarios, leading to improved performance and broader applicability (Sastry et al., 2023; Kim et al., 2022).

## 4.6 Ablation Experiment

To demonstrate the effectiveness of our approach, we conduct ablation studies on the "Suicide-IED" dataset. (1) **IGE Module Ablation Experiment:** To prove the effectiveness of our approach, we conduct experiments as shown in Table 3. JSON is a prevalent format for representing structured data. We encode the data in JSON format and instruct the LLMs to perform expansion, while maintaining the rest of the process consistent with the Python prompt approach. As shown in the Table, the results obtained through the use of Python prompts are noticeably better than those achieved with JSON prompts. And after filtering, the event types generated by the Python prompt are significantly more numerous than those generated by the JSON prompt. This observation underscores the effectiveness of the Python prompt approach. (2) **Diffusion Model Ablation Experiment:** In Table 4, comparing our **KDM** model with a variant

| Model | Event type Match(F1) | Event sequence match(F1) | | KL divergence | | EN |
|---|---|---|---|---|---|---|
| | | l=2 | l=3 | Node-type | Edge-type | |
| without IGE | 0.672 | 0.411 | 0.216 | 1.96 | 3.67 | - |
| IGE with JSON prompt | 0.685 | 0.418 | 0.221 | 1.94 | 3.89 | 2 |
| IGE with Python prompt | **0.713** | **0.462** | **0.268** | **1.91** | **3.45** | **9** |

Table 3: Results of different prompts for the IGE Module on Suicide-IED dataset. **EN** is the number of effective events generated by the LLM after filtering, which are used for Instance Graph Expansion.

| Model | Event type Match(F1) | Event sequence match(F1) | | KL divergence | |
|---|---|---|---|---|---|
| | | l=2 | l=3 | Node-type | Edge-type |
| **KDM** | **0.713** | **0.462** | **0.268** | **1.91** | **3.45** |
| w/o depth | 0.709 | 0.429 | 0.209 | 1.92 | 3.83 |
| w/o IGE | 0.672 | 0.411 | 0.216 | 1.96 | 3.67 |

Table 4: The diffusion model ablation experiment on Suicide-IED dataset. "w/o depth" denoted as trained the model without depth features in graph transformer; "w/o IGE" denoted as trained the model on the dataset without Instance Graph Expansion module.

that removes the Instance Graph Expansion module, Our model achieves a 4.1% increase in node matching accuracy, proving the effectiveness of Instance Graph Expansion module. Additionally, by incorporating depth information, we observe a notable 5.9% improvement in sequence matching. These results demonstrate that the inclusion of depth information enhances our model's ability to capture the structural characteristics of graph, proving the effectiveness of adding depth features. (3) **Entity Predictor Ablation Experiment:** In Figure 4, Compared to not setting weight hyperparameters, our model achieves a significant 5.57% improvement in the macro F1 index, demonstrating that our weight scalar function significantly addresses the long-tail data problem. Moreover, when comparing the results of "w/o RE" and "w/o IGE", the improvements highlight the effectiveness of adding role and event features and Instance Graph Expansion module.

## 4.7 Case Study

In Figure 5, we observe that our Instance Graph Expansion module successfully generates a schema that encompasses a broader range of events and exhibits more comprehensive temporal relationships within complex events. This outcome support the effectiveness of leveraging object-oriented coding to distill knowledge from LLMs. Additionally, we provide a case study showcasing the diffusion process on the "Suicide-IED" dataset in Figure 6 in Appendix.

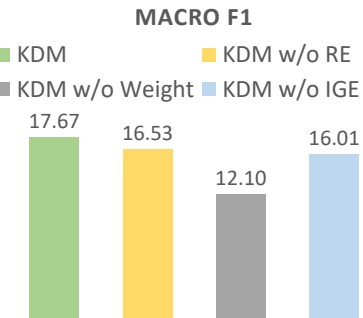

Figure 4: The entity predictor ablation experiment. "w/o RE" denoted as trained without fixed role embedding features and event embedding features; "w/o weight" denoted as trained without hyperparameter weight; "w/o IGE" denoted as trained without Instance Graph Expansion module.

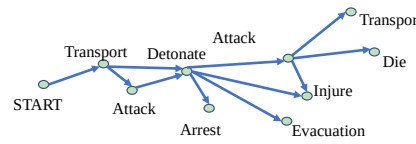

(a) the event schema skeleton without Instance Graph Expansion module

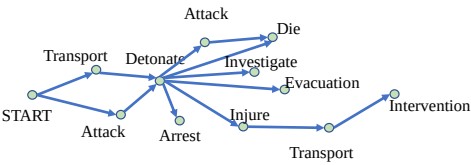

(b) the event schema skeleton with Instance Graph Expansion module

Figure 5: Case Study on event schema skeleton

## 5 Related Work

**Event Schema**   Event schema induction is a comprehensive graphical pattern composed of temporal and multi hop argument relationships (Li et al., 2020; Jin et al., 2022). It is actually a combination of atomic schema induction (Chambers, 2013; Yuan et al., 2018; Du and Ji, 2022; Wang et al., 2021) and script learning (Rudinger et al., 2015; Granroth-Wilding and Clark, 2016; Weber et al., 2018). Clearly, the event schema induction has broad application significance. For example, the event schema facilitates analysis and prediction of future events, aiding in the development of reaction plans for relevant scenarios (Li et al., 2021; Dror et al., 2023; Pan et al., 2021). Event schemas can be used as guidance information in information extraction, which helps people understand the internal logic of events (Wen et al., 2021).

**Diffusion models**   Diffusion model (Sohl-Dickstein et al., 2015; Ho et al., 2020) has achieved impressive results on image, text and audio generation (Rombach et al., 2022; Shen et al., 2023; Li et al., 2022; Gong et al., 2022; Kong et al., 2020; Yuan et al., 2022). Recently, Vignac et al. (2022) have shown great potential in graph generation field. Previous graph diffusion models embedded graphs in a continuous space by adding Gaussian noise to the nodes and edges feature (Niu et al., 2020; Jo et al., 2022). However, this approach destroys the graph's sparsity and makes it hard to capture the node connections (Vignac et al., 2022). Discrete diffusion model (Austin et al., 2021; Yang et al., 2023; Vignac et al., 2022; Johnson et al., 2021) overcomes this problem by utilizing Markov process that can occur independently on each node or edge.

## 6 Conclusion

In this work, we identify the limitations of previous works and proposed a Knowledge-enriched discrete diffusion model. To enhance the quality and coherence of the generated schemas, we harness the potential and rich knowledge present in LLMs by utilizing them for Instance Graph Expansion. Our model leverages a discrete diffusion process to learn and generate event skeletons, while incorporating an entity relationship predictor to predict the relationships between event arguments. Additionally, we propose a conditional diffusion model with the purpose of generating schemas for multiple diverse topics. We achieved the best results among multiple different evaluation indicators.

## Limitations

We only consider the temporal relationship between events here and do not consider the hierarchical structure of the event schema, which may result in not perfect event schemas generated by us.

Due to the limited availability of datasets, our conditional diffusion model **KDMall** has only undergone unified training and testing on three highly related explosive events, requiring more categories and quantities of data for the comprehensive ability testing of the model.

## Ethics Statement

We use a discrete diffusion model to generate event skeletons and design an entity relationship predictor. At the same time, we have fully explored the potential rich knowledge in LLM for knowledge expansion. Our work has improved the effectiveness of event schema induction, helping people better summarize the logic and ontology knowledge of events, making contributions to this field.

## Acknowledgements

This work is supported by the National Key Research and Development Program of China (No. 2020AAA0106400), the National Natural Science Foundation of China (No. 62176257, 61976211). This work is also supported by the Strategic Priority Research Program of Chinese Academy of Sciences (Grant No.XDA27020100 ), the Youth Innovation Promotion Association CAS, and Yunnan Provincial Major Science and Technology Special Plan Projects (No.202202AD080004).

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

## A    Data Preprocessing

In the data preprocessing stage. Firstly, for each complex event, we constructed an instance graph by merging coreferential events or entities. Isolated events were excluded from the instance graphs during the graph construction process.

Specifically, we followed the cleaning strategy outlined in Jin et al. (2022). We deleted links with the same start and end types, as well as event-event links such as (DIE, INJURE), (ARRESTJAILDETAIN, ATTACK), (ENDPOSITION, STARTPOSITION), (DEFEAT, EXCHANGEBUYSELL), (SENTENCE, DIE), (ENDPOSITION, SENTENCE), and (THREATENCOERCE, RELEASEPAROLE) from the instance graphs. The maximum number of graph nodes m is set to 50.

## B    Training And Evaluation Details

In our event skeleton induction process, we utilize a 12-layer Transformer model. Additionally, we employ a 3-layer Transformer as our entity relation predictor. To balance the trade-off between nodes and edges, we set $\lambda$ to 3. The learning rate is set to 1e-4, and the number of diffusion training epochs is set to 2500. The scalar function $H(\hat{r}_{ij})$ is set to 0.1 if $r_{ij}$ indicates "No-Relation", otherwise the function is set to 0.9. We conduct evaluations using 500 randomly generated event schemas for each performance metric. The node number is sampled from a range of 25 to 35. We choose the model checkpoint from the last epoch for evaluation.

In the Instance Graph Expansion process, we select the top 10 frequently occurring event sequences from the training data as inputs for ChatGPT. Each event sequence is input to ChatGPT 10 times to obtain the final result. Furthermore, we use a hyperparameter K of 3 to filter out events generated by ChatGPT that occur less frequently.

In the Entity Relation Prediction module, each event has a predetermined set of argument roles. For example, the "Injure" event may have the argument role "Victim" limited to entity types "PER" and "AML". We count the occurrences of entity categories for each role in all instance graphs. The entity category with the highest occurrence in the corresponding role is then inserted into the event skeleton.

To modify the test data, we made the following modifications: 1. Merge the same path: For all subsequent nodes of each event node, merge event

nodes with the same type, starting from the START node and merging in the order of the BFS algorithm. 2. Supplementary event nodes: Based on human judgment, randomly add possible missing events that may occur in the schema.

## C Conditional Discrete Diffusion Model

**Transition probability matrix in forward process.** In discrete diffusion model, a transition probability matrix $\boldsymbol{Q}$ is defined to corrupt data for each step. Here $\boldsymbol{Q}_t = \alpha^t \mathbf{I} + (1 - \alpha^t)\mathbf{1}(\mathbf{1})^T/K$, where $\mathbf{1}$ is a column vector of all ones, $\alpha^t$ varies from 1 to 0 making sure the node and edge feature sampled from is a uniform distribution at time $T$ (Hoogeboom et al., 2021; Yang et al., 2023). $\beta^t = (1 - \alpha^t)/K$ and the transition matrix can be represent as:

$$
\boldsymbol{Q}_t = \begin{bmatrix} \alpha^t + \beta^t & \beta^t & \dots & \beta^t \\ \beta^t & \alpha^t + \beta^t & \dots & \beta^t \\ \cdot & \cdot & \dots & \cdot \\ \cdot & \cdot & \dots & \cdot \\ \beta^t & \beta^t & \dots & \alpha^t + \beta^t \end{bmatrix}
$$

we can calculate $q(\boldsymbol{x}_t|\boldsymbol{x}_0)$ according to following formula:

$$
\boldsymbol{x}_0\bar{\boldsymbol{Q}}_t = \bar{\alpha}^t\boldsymbol{x}_0 + \bar{\beta}^t. \tag{20}
$$

as $t$ is enough large, $\alpha^t$ is close to 0, the graph distribution $\mathbf{G}_t$ is confirm to uniform distribution.

**Reverse discrete diffusion process**, we convert the noise $\boldsymbol{G}_T$ into $\boldsymbol{G}$, whose joint probability having a Markovian structure follows (Vignac et al., 2022):

$$
\log p_\theta(\boldsymbol{G}_{0:T}) = \log p(\boldsymbol{G}_T) \sum_{t=1}^{T} \log p_\theta(\boldsymbol{G}_{t-1}|\boldsymbol{G}_t)
$$

where $p_\theta$ is the process of the reverse with learnable parameters $\theta$. and for each discrete elements $x$ in graph $\boldsymbol{G}_{0:T}$ posterior probability is :

$$
p_\theta(x_{t-1}|x_t) = \sum_x q(x_{t-1}|x_t, x)p(x)
$$
$$
= \sum_{x_0} q(x_{t-1}|x_t, x_0)p_\theta(x_0|x_t)
$$

The posterior $q(x_{t-1}|x_t, x_0)$ can be derived according to Bayes formula as follows (Austin et al.,

2021):

$$
q(x_{t-1}|x_t, x_0) = \frac{q(x_t|x_{t-1}, x_0)q(x_{t-1}|x_0)}{q(x_t|x_0)}
$$
$$
= \frac{x_t(\boldsymbol{Q}_t)^T \odot x_0\bar{\boldsymbol{Q}}_{t-1}}{x_0\bar{\boldsymbol{Q}}_t(x_t)^T}
$$

To train the discrete diffusion process, we minimize the negative logarithmic likelihood of the predicted distribution of the model using variational lower bound (VLB), We use $\boldsymbol{G}_0$ here to represent $\boldsymbol{G}$:

$$
\begin{aligned}
\mathcal{L} &= -E_{q(\boldsymbol{G}_0)}[log p_\theta(\boldsymbol{G}_0)] \\
&\leq \mathcal{L}_{VLB} \\
&= -E_{q(\boldsymbol{G}_{0:G})}[\log p_\theta(\boldsymbol{G}_0)] \\
&= \mathbb{E}_{q(\boldsymbol{G}_{0:T})}[\underbrace{D_{KL}(q(\boldsymbol{G}_T|\boldsymbol{G}_0) \parallel p_\theta(\boldsymbol{G}_T))}_{L_T} \\
&\quad + \sum_{t=2}^{T} \underbrace{D_{KL}(q(\boldsymbol{G}_{t-1}|\boldsymbol{G}_t, \boldsymbol{G}_0) \parallel p_\theta(\boldsymbol{G}_{t-1}|\boldsymbol{G}_t))}_{L_{t-1}} \\
&\quad \underbrace{- \log p_\theta(\boldsymbol{G}_0|\boldsymbol{G}_1)]}_{L_0} \\
&= \mathbb{E}_{q(\boldsymbol{G}_{0:T})}[\underbrace{D_{KL}(q(\boldsymbol{G}_T|\boldsymbol{G}_0) \parallel p_\theta(\boldsymbol{G}_T|\boldsymbol{G}_n, \boldsymbol{G}_d))}_{L_T} \\
&\quad + \log p_\theta(\boldsymbol{G}_n, \boldsymbol{G}_d) \\
&\quad + \sum_{t=2}^{T} \underbrace{D_{KL}(q(\boldsymbol{G}_{t-1}|\boldsymbol{G}_t, \boldsymbol{G}_0) \parallel p_\theta(\boldsymbol{G}_{t-1}|\boldsymbol{G}_t))}_{L_{t-1}} \\
&\quad \underbrace{- \log p_\theta(\boldsymbol{G}_0|\boldsymbol{G}_1)]}_{L_0}
\end{aligned}
$$

Please note that $\boldsymbol{G}_t$ is sampled from the node number distribution $\boldsymbol{G}_n$ and the corresponding depth distribution $\boldsymbol{G}_d$. Therefore, the probability $p_\theta(\boldsymbol{G}_T)$ can be expressed as $p_\theta(\boldsymbol{G}_T) = p_\theta(\boldsymbol{G}_T|\boldsymbol{G}_n, \boldsymbol{G}_d)p_\theta(\boldsymbol{G}_n, \boldsymbol{G}_d)$.

The terms $L_T$ and $L_{t-1}$ represent the Kullback-Leibler (KL) divergences between graph categorical distributions, while $L_0$ represent the predicted probabilities of the graph $\boldsymbol{G}_0$ based on the noisy graph $\boldsymbol{G}_1$. The algorithms 1 and 2 is the training and generating algorithms about **KDM**.

## D Supplement Experiment

As presented in Table 5, we evaluate our model on the original testing data used by (Jin et al., 2022) and observe consistent outperformance of our model compared to the baselines specially in sequence match. This result highlights the strong capability of our discrete diffusion model in generating high-quality event schemas. Interestingly, we

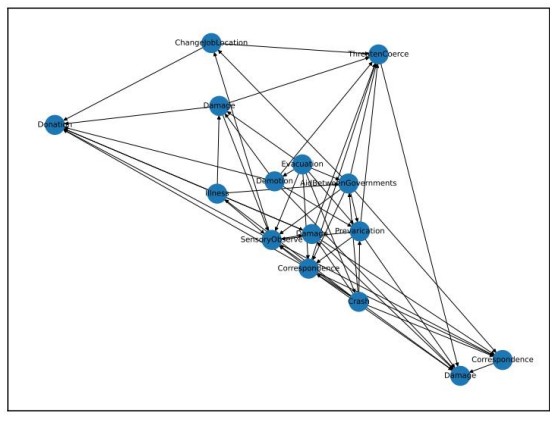

(a) t=500

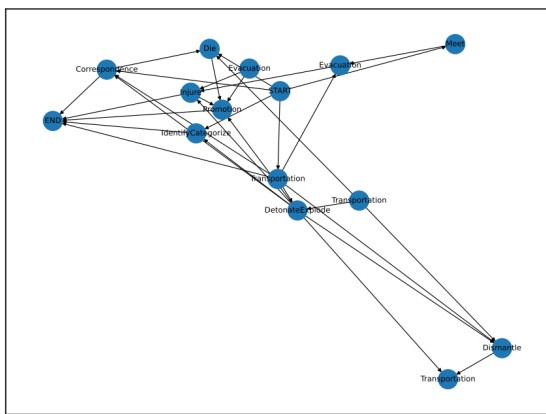

(b) t=250

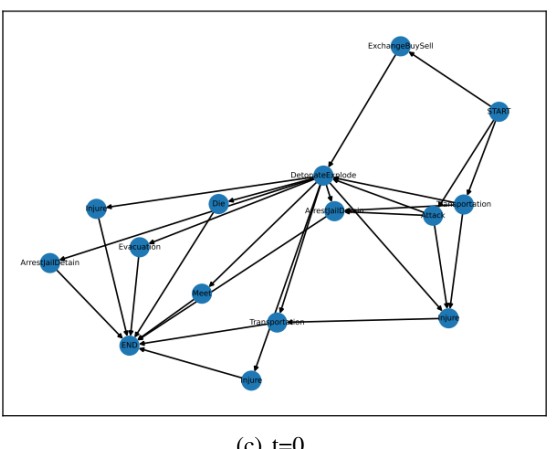

(c) t=0

Figure 6: Case Study on diffusion process. During the reverse process, t gradually changes from 500 to 0, and the corresponding schema for t at different times corresponds to the above.

note that models without knowledge expansion exhibited better performance in this evaluation. This finding also suggests that the original test data may not effectively measure the quality of the generated schemas.

---

**Algorithm 1:** Training **KDM**

---

1 **for** $G$ *in* $\mathcal{D}$ **do**
2      sample $t$ from Uniform(T);
3      sample $\mathcal{G}_t$ from distribution
        $(\boldsymbol{N}\bar{\boldsymbol{Q}}_t^N, \boldsymbol{E}\bar{\boldsymbol{Q}}_t^E)$;
4      estimate distribution $\boldsymbol{G}_p^t = \phi_\theta(\mathcal{G}_t)$;
5      calculate loss $\mathcal{L}_{CE}(\boldsymbol{G}_p^t, \boldsymbol{G})$;
6      update network $\phi_\theta$
7 **end**

---

**Algorithm 2:** Sampling from **KDM**

---

1 sample $\mathcal{G}_T$ from Uniform distribution;
2 **for** $t = T$ *to 1* **do**
3      convert $\mathcal{G}_t$ to distribution $\boldsymbol{G}_t$;
4      $\boldsymbol{G}_p^t = \phi_\theta(\boldsymbol{G}_t)$;
5      estimate distribution $p_\theta(\boldsymbol{G}_{t-1}|\boldsymbol{G}_t)$;
6      sample $\mathcal{G}_{t-1}$ from distribution;
7 **end**

---

## E    Instance Graph Expansion Details

The process of the Instance Graph Expansion is shown in Figure 7, we also present our event knowledge expansion prompt in Figure 8, temporal relation expansion prompt in Figure 10 and entity relation expansion prompt in Figure 9.

| Dataset | Model | Event type Match(F1) | Event sequence match(F1) l=2 | l=3 | KL divergence Node-type | Edge-type |
|---|---|---|---|---|---|---|
| General-IED | FBS | 0.617 | 0.149 | 0.064 | 1.88 | 4.32 |
| | DoubleGAE | 0.697 | 0.273 | 0.128 | **1.66** | 4.96 |
| | **KDM** | 0.663 | 0.318 | 0.132 | 3.01 | 5.08 |
| | **KDM** w/o IGE | **0.698** | **0.327** | **0.138** | 2.52 | **4.87** |
| Car-IED | FBS | 0.542 | 0.126 | 0.038 | 4.12 | 6.37 |
| | DoubleGAE | 0.674 | 0.259 | 0.081 | 2.14 | 5.42 |
| | **KDM** | 0.702 | 0.371 | 0.180 | 2.07 | 4.47 |
| | **KDM** w/o IGE | **0.757** | **0.411** | **0.185** | **1.42** | **4.24** |
| Suicide-IED | FBS | 0.642 | 0.164 | 0.048 | 2.39 | 6.36 |
| | DoubleGAE | 0.709 | 0.290 | 0.095 | 1.76 | 5.91 |
| | **KDM** | 0.688 | 0.415 | 0.217 | 2.13 | 3.75 |
| | **KDM** w/o IGE | **0.738** | **0.467** | **0.268** | **1.61** | **3.50** |

Table 5: Schema matching score (%) by checking the intersection of induced schemas and test schemas. both the training and testing instances are not through Knowledge Expansion. The baseline's results are provided by previous work.

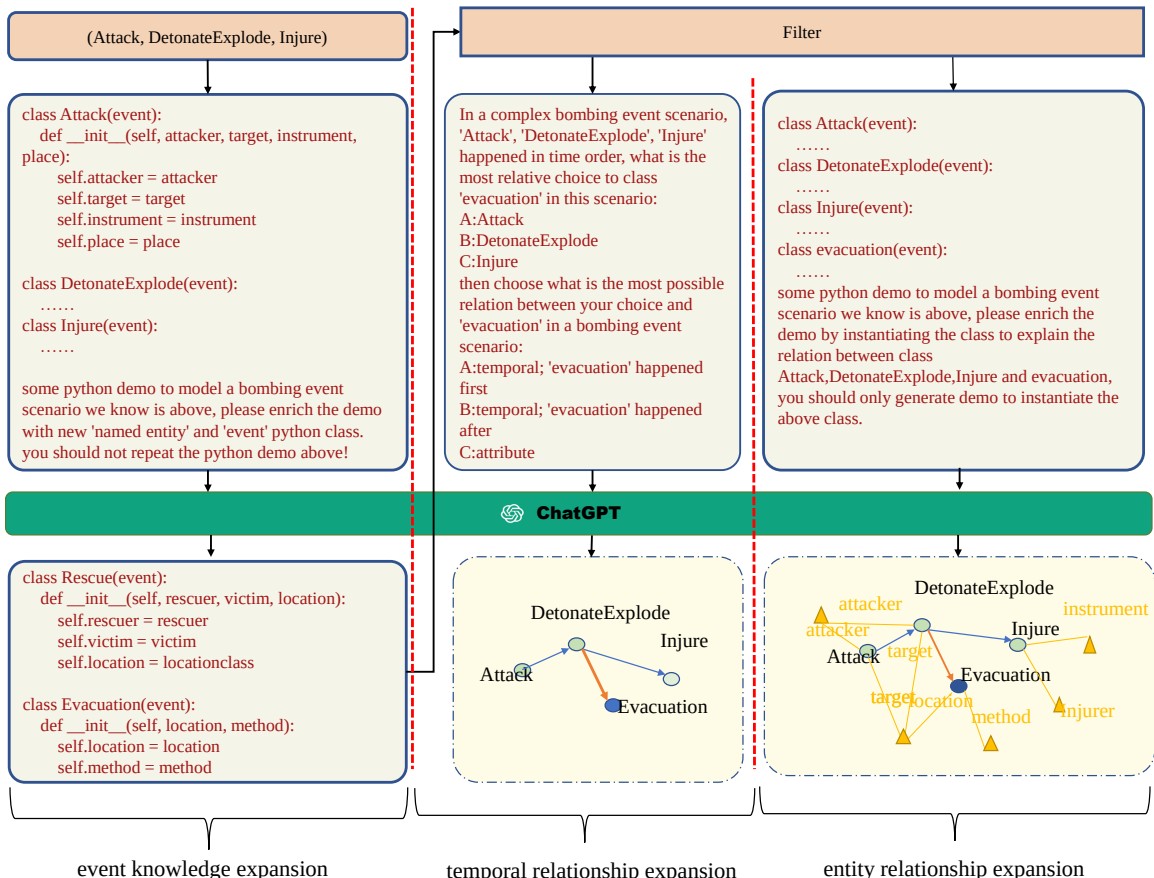

Figure 7: This figure illustrates the Instance Graph Expansion process. The event knowledge expansion process write selected event sequences into Python code and complete them by ChatGPT to generate new events; The temporal relationship expansion process utilize choice question to link new event to sequence; The entity relationship expansion process instantize the event class to get argument relations.

**Input:**

```
class Attack(event):
    def __init__(self, attacker, target, instrument, place):
        self.attacker = attacker
        self.target = target
        self.instrument = instrument
        self.place = place
class DetonateExplode(event):
    def __init__(self, attacker, target, instrument, explosive_device, place):
        self.attacker = attacker
        self.target = target
        self.instrument = instrument
        self.explosive_device = explosive_device
        self.place = place
class Injure(event):
    def __init__(self, victim, injurer, instrument):
        self.victim = victim
        self.injurer = injurer
        self.instrument = instrument
```

some python demo to model a bombing event scenario we know is above, please enrich the demo with new 'named entity' and 'event' python class. you should not repeat the python demo above!you should not repeat the python demo above!you should not repeat the python demo above!

**Answer:**

```
class Bombing(event):
    def __init__(self, attacker, target, instrument, place, casualties):
        self.attacker = attacker
        self.target = target
        self.instrument = instrument
        self.place = place
        self.casualties = casualties
class Evacuation(event):
    def __init__(self, location, evacuees, reason):
        self.location = location
        self.evacuees = evacuees
        self.reason = reason
class Investigation(event):
    def __init__(self, location, investigators, start_date):
        self.location = location
        self.investigators = investigators
        self.start_date = start_date
```

Figure 8: Instance of event knowledge expansion process

**Input:**

In a complex bombing event scenario, 'Attack', 'DetonateExplode', 'Injure' happened in time order, what is the most relative choice to class 'evacuation' in this scenario, please choose your answer from:
A:Attack
B:DetonateExplode
C:Injure

then choose what is the most possible relation between your choice and 'evacuation' in a bombing event scenario, please choose your answer from:
A:temporal; 'evacuation' happened first
B:temporal; 'evacuation' happened after
C:attribute
(attention you can only answer the above two questions only with A or B or C)

**Answer:**

B: DetonateExplode
B: Temporal; 'evacuation' happened after

Figure 9: Instance of temporal relation expansion process

**Input:**

```
class Attack(event):
    def __init__(self, attacker, target, instrument, place):
        self.attacker = attacker
        self.target = target
        self.instrument = instrument
        self.place = place
class DetonateExplode(event):
    def __init__(self, attacker, target, instrument, explosive_device, place):
        self.attacker = attacker
        self.target = target
        self.instrument = instrument
        self.place = place
class Injure(event):
    def __init__(self, victim, injurer, instrument):
        self.victim = victim
        self.injurer = injurer
        self.instrument = instrument
class Evacuation(event):
    def __init__(self, location, method):
        self.location = location
        self.method = method
```

some python demo to model a bombing event scenario we know is above, please enrich the demo by instantiating the class to explain the relation between class Attack,DetonateExplode,Injure and Bombing, you should only generate demo to instantiate the above class.

one demo example is below:

```
# Instantiate Transport event
transport_event = Transport(Vehicle ="car", Origin ="park", Destination ="Building)
# Instantiate DetonateExplode event
detonate_explode_event = DetonateExplode(attacker="John", target="Building", instrument="Explosive",
explosive_device="C4", place="City Center")
# Instantiate Investigate event
investigate_event = Investigate(location="City Center", investigator="Detective Smith")
```

**Answer:**

```
# Instantiate Attack event
attack_event = Attack(attacker="John", target="Building", instrument="Explosive", place="City Center")
# Instantiate DetonateExplode event
detonate_explode_event = DetonateExplode(attacker="John", target="Building", instrument="Explosive",
explosive_device="C4", place="City Center")
# Instantiate Injure event
injure_event = Injure(victim="Jane", injurer="John", instrument="Explosive")
# Instantiate Evacuation event
evacuation_event = Evacuation(location="City Center", method="Emergency evacuation")
```

Figure 10: Instance of entity relation expansion process