# OpenReview forum: "Complex Event Schema Induction with Knowledge-Enriched Diffusion Model"
_EMNLP/2023/Conference — EMNLP 2023 Findings_

### Official Review · Reviewer_VM43 · 2023-08-03

**Soundness:** 2

**Excitement:**

3: Ambivalent: It has merits (e.g., it reports state-of-the-art results, the idea is nice), but there are key weaknesses (e.g., it describes incremental work), and it can significantly benefit from another round of revision. However, I won't object to accepting it if my co-reviewers champion it.

**Paper Topic And Main Contributions:**

This paper addresses the problem of inducing event schemas that are more complex than atomic event representations (eg. event tuples) and may have correlations, which is an important problem. The authors propose an improved method to address two major issues in existing schema induction approaches i) below-par coverage of instance graphs ii) error propagation in auto-regressive schema generation. The key ideas to resolve the issues are respectively i) use zero-shot ChatGPT prompting to extract related entities and events, and expand the instance graphs ii) employ non auto-regressive approach based on discrete diffusion model to predict the schema graph nodes and edges. Experiments have been done on 3 datasets for performance analysis in terms of F1-scores.

**Questions For The Authors:**

1. What different ChatGPT prompting formats were explored, and why is the adopted prompting better?
2. What are the sizes of the datasets in Sec 4?

**Reasons To Accept:**

- Complex event schema induction is an important problem.
- Specific ideas to address key issues with existing approaches, and propose an improved method.
- Better performance reported in empirical studies.

**Reasons To Reject:**

Though the proposed method has been demonstrated empirically, I think the paper is not very sound.

- Problem formulation (Sec 2) is confusing and unclear. It starts by defining the diffusion model which I think should come later. Sec 2.2 is really the problem formulation but this is not at all sufficient. It does not formulate  the schema induction problem, nor define the concepts involved. Without this, I think it is difficult to understand the contributions in the rest of the paper.
- Sec 4 Experiments need lot more details, eg. what are the sizes of the datasets. Also, given that 'we manually modify the test data, generating golden text event schemas based on the modified data', the unmodified results should also be included. Also I think the modified data should be released publicly.
- In Sec 4.3, the metrics are defined rather informally. In addition to F1-measure, I would like to see precision and recall numbers, to check if there is any tradeoffs.
- ChatGPT prompting methodology is not explained well. What different formats were explored, and why is the adopted prompting better?
- Sec 4.7 case study is too brief to be worthwhile.
- I think there's too much space devoted for covering the theory behind diffusion model, but much less space for explaining the main problem statement, technical contributions and evaluation details.  This paper needs major revisions to address this.

**Reproducibility:**

3: Could reproduce the results with some difficulty. The settings of parameters are underspecified or subjectively determined; the training/evaluation data are not widely available.

**Reviewer Confidence:**

2: Willing to defend my evaluation, but it is fairly likely that I missed some details, didn't understand some central points, or can't be sure about the novelty of the work.

**Typos Grammar Style And Presentation Improvements:**

I found several grammar issues, eg.

- [Page 2] 'Therefore, how to improve the knowledge coverage of instance graphs is an important problem.'
- [Page 2] "..predicts all nodes and links simultaneously in non-auto-regressive manner but not generates individually based on time series"

---

> ### Author Rebuttal · Authors · 2023-08-28
>
> Thanks for your careful and insightful reviews.
>
> **Response to Rejection 1: Problem formulation (Sec 2) is confusing and unclear.Sec 2.2 is really the problem formulation but this is not at all sufficient. It does not formulate the schema induction problem, nor define the concepts involved.....**
>
> Thank you very much for your suggestions. We will make adjustments to the placement of the definition section of the diffusion model.
> As for the Problem formulation, in lines 180 to 196 of our paper, particularly in lines 193 to 196, we provided a formal and conceptual definition of our problem.
>
> **Response to Rejection 2: -   Sec 4 Experiments need lot more details, eg. what are the sizes of the datasets. Also, given that 'we manually modify the test data, generating golden text event schemas based on the modified data', the unmodified results should also be included......**
>
> I apologize that we do not provide detailed information about the dataset. The training data sizes in the IED dataset are as follows: suicide\_ied:105, car\_bombing\_ied:38, general\_ied:59.
>
> In the main text (i.e., Section 4), we conduct experiments using modified data. To ensure a fair and comprehensive comparison, we include supplementary experiments in the Appendix D. The results of evaluating on unmodified data are presented in Table 4, and the analysis of these results is provided between Line 904 and Line 916.
>
> We will make our modified data and the source code publicly available for reproducibility and further research.
>
> **Response to Rejection 3: the metrics are defined rather informally. In addition to F1-measure, I would like to see precision and recall numbers, to check if there is any tradeoffs.**
>
> I apologize for the oversight. Actually, our evaluation metrics are in line with previous studies. Due to space limitations, precision and recall are not included in the presentation of our experimental results. However, we have provided a more comprehensive set of outcomes, as illustrated in the Table below.
>
> | Dataset     | Event type match prec | Event type match rec | seq match(l=2) prec | seq match(l=2) rec | seq match(l=3) prec | seq match(l=3) rec |
> | ----------- | --------------------- | -------------------- | ------------------- | ------------------ | ------------------- | ------------------ |
> | General-IED | 0.738                 | 0.739                | 0.449               | 0.438              | 0.233               | 0.228              |
> | Car-IED     | 0.681                 | 0.751                | 0.374               | 0.452              | 0.193               | 0.267              |
> | Suicide-IED | 0.703                 | 0.765                | 0.462               | 0.531              | 0.279               | 0.350              |
>
> We observe that our model's recall is higher than the precision. We analyze that there could be several reasons for this disparity:
>
> (1) The discrete diffusion model relies on probability sampling, which introduces a chance of selecting nodes and edges with relatively lower probabilities. This phenomenon can lead to a higher recall, as it encompasses a broader range of possibilities.
>
> (2) The training dataset includes a more diverse set of sample instances, whereas the testing dataset is comparatively limited. Consequently, certain scenarios present in the training phase might not be adequately represented during testing, contributing to the observed difference.
>
> **Response to Rejection 4:  ChatGPT prompting methodology is not explained well. What different formats were explored, and why is the adopted prompting better?**
>
> I apologize for not adequately conveying the details and innovative aspects of our IGE module.
>
> Complex event schemas involve intricate graph structures and structured knowledge representation. LLMs are good at processing unstructured language tasks, but there isn't a well-established method to make structured graph schema understandable to LLMs. When we had LLMs do IGE directly, they generated extensive unstructured natural language text. While this text contain knowledge, converting it into usable structured knowledge that we can use is challenging. To solve this problem we need LLMS to be able to handle structured inputs and outputs.
>
> To retain structured instance graph information and use LLMs' coding power, we treated events as Python objects. We recognize that events, entities, and their intricate relationships can correspond, respectively, to classes, attributes, and instances in the object-oriented paradigm. To prove our approach works, we designed comparative experiments as shown in the table below.
>
>
> | IGE module             | Event type match(F1) | Event sequence match(l=2) | Event sequence match(l=3) | KL divergence(Node type) | KL divergence(Edge type) | EN    |
> | :--------------------- | -------------------- | ------------------------- | ------------------------- | ------------------------ | ------------------------ | ----- |
> | without IGE            | 0.672                | 0.411                     | 0.216                     | 1.96                     | 3.67                     | -     |
> | IGE with JSON prompt   | 0.685                | 0.418                     | 0.221                     | 1.94                     | 3.89                     | 2     |
> | IGE with Python prompt | **0.713**            | **0.462**                 | **0.268**                 | **1.91**                 | **3.45**                 | **9** |
>
>
>
> The table express the results of different prompts for the IGE Module on Suicide-IED dataset. **EN** is the number of effective events generated by the LLM after filtering, which are used for instance graph expansion.
>
> JSON is a prevalent format for representing structured data. We encoded the data in JSON format and instructed the large model to perform expansion, while maintaining the rest of the process consistent with the Python prompt approach. As shown in the Table, the results obtained through the use of Python prompts are noticeably better than those achieved with JSON prompts. And after filtering, the event types generated by the Python prompt are significantly more numerous than those generated by the JSON prompt. This observation underscores the effectiveness of the Python prompt approach.
>
> **Response to Rejection 5:  Sec 4.7 case study is too brief to be worthwhile.**
>
> The "Sec 4.7 Case Study" is included to showcase the effectiveness of our IGE module. This segment aim to provide a more intuitive illustration of how event schemas that undergo knowledge expansion through LLMs are not only more efficient but also more refined, thereby emphasizing the validity of our IGE module.
>
> **Response to Rejection 6: I think there's too much space devoted for covering the theory behind diffusion model, but much less space for explaining the main problem statement, technical contributions and evaluation details.**
>
> Recognizing that the diffusion model serves as a fundamental pillar of our approach, we acknowledge the significance of clarifying its operational mechanisms to enhance the understanding of our contributions and innovations.  Accordingly, we have allocated a substantial portion of our work to comprehensively expound on the theory underpinning the diffusion model, and this content is mainly housed within the appendix.
>
> We have delineated the principal challenges in Introduction sections spanning Line 70 to Line 87 and Line 88 to Line 107, while also providing a more visually intuitive representation of the task through Figure 1.  A concise encapsulation of our technical contributions can be found within Line 135 to Line 150.  Furthermore, detailed elucidation of training and evaluation specifics has been presented in Line 420 to Line 444 and Line 828 to Line 857.
>
> **Response to Question A: What different ChatGPT prompting formats were explored, and why is the adopted prompting better?**
>
> Please see the response to Rejection 4.
>
> **Response to Question B: What are the sizes of the datasets in Sec 4?**
>
> Please see the response to Rejection 2.
>
> **Response to Typos Grammar Style And Presentation Improvements**
>
> Thank you very much for pointing out the spelling, grammar, and expression issues. We will correct these errors in the revised version.

---

### Official Review · Reviewer_WC1S · 2023-08-04

**Soundness:** 4

**Excitement:**

3: Ambivalent: It has merits (e.g., it reports state-of-the-art results, the idea is nice), but there are key weaknesses (e.g., it describes incremental work), and it can significantly benefit from another round of revision. However, I won't object to accepting it if my co-reviewers champion it.

**Paper Topic And Main Contributions:**

The motivation the paper is to overcome the challenges such as error propagation and data quality issues in event schema induction. Existing methods just induce the atomic schema without considering correlations between events and struggle with extracting complete instance information.

The paper proposes a knowledge-enriched discrete diffusion model for complex event schema induction task. They mainly propose an instance graph expansion module to improve knowledge coverage of instance graphs and an event skeleton generation module which predict all nodes and links altogether.

**Questions For The Authors:**

Have you designed any experiment to inspect and validate auto-generated codes from LLM?

**Reasons To Accept:**

The motivation and contribution of the paper is clear.

Various experiments and analysis are carried out on a hot topic.

**Reasons To Reject:**

Authors need to elaborate their ablation study regarding instance graph expansion module. Even though authors presented some discussions in table 3 and in section 4.7, they did not tackle the problem of reliability and robustness of such auto-generated code snippets from LLMs.

Authors' experiments were conducted on just one specific event scenario. They should have evaluated their methods on a set of different event scenario such as disease outbreak, coup, riots or any others. It's hard to say the proposed method is superior to the existing methods, seeing the performance on a single scenario.


**Reproducibility:**

4: Could mostly reproduce the results, but there may be some variation because of sample variance or minor variations in their interpretation of the protocol or method.

**Reviewer Confidence:**

5: Positive that my evaluation is correct. I read the paper very carefully and I am very familiar with related work.

---

> ### Author Rebuttal · Authors · 2023-08-28
>
> Thanks for your careful and insightful reviews. We will utilize these insights as a foundational framework to augment and refine our works. Here, we've outlined a detailed response addressing each of your feedback points:
>
> **Response to Rejection 1: Authors need to elaborate their ablation study regarding instance graph expansion module. The article did not tackle the problem of reliability and robustness of such auto-generated code snippets from LLMs.**
>
> I apologize for not adequately conveying the details and innovative aspects of our IGE (instance graph expansion) module. Based on your advice, we've added the following explanation:
>
> Complex event schemas involve intricate graph structures and structured knowledge representation. LLMs are good at processing unstructured language tasks, but there isn't a well-established method to make structured graph schema understandable to LLMs. When we had LLMs do IGE directly, they generated extensive unstructured natural language text. While this text contain knowledge, converting it into usable structured knowledge that we can use is challenging. To solve this problem we need LLMS to be able to handle structured inputs and outputs.
>
> To retain structured instance graph information and use LLMs' coding power, we treated events as Python objects. We recognize that events, entities, and their intricate relationships can correspond, respectively, to classes, attributes, and instances in the object-oriented paradigm.
>
> To better elucidate the reliability of the LLM-generated code approach, we devised experiments as illustrated in the table below.
>
>
>
> | IGE module             | Event type match(F1) | Event sequence match(l=2) | Event sequence match(l=3) | KL divergence(Node type) | KL divergence(Edge type) | EN    |
> | ---------------------- | -------------------- | ------------------------- | ------------------------- | ------------------------ | ------------------------ | ----- |
> | without IGE            | 0.672                | 0.411                     | 0.216                     | 1.96                     | 3.67                     | -     |
> | IGE with JSON prompt   | 0.685                | 0.418                     | 0.221                     | 1.94                     | 3.89                     | 2     |
> | IGE with Python prompt | **0.713**            | **0.462**                 | **0.268**                 | **1.91**                 | **3.45**                 | **9** |
>
> The table express the results of different prompts for the IGE Module on Suicide-IED dataset. **EN** is the number of effective events generated by the LLM after filtering, which are used for instance graph expansion.
>
> JSON is a prevalent format for representing structured data. We encoded the data in JSON format and instructed the large model to perform expansion, while maintaining the rest of the process consistent with the Python prompt approach.
>
> As shown in the Table, the results obtained through the use of Python prompts are noticeably better than those achieved with JSON prompts. And after filtering, the event types generated by the Python prompt are significantly more numerous than those generated by the JSON prompt. This observation underscores the effectiveness of the Python prompt approach.
>
> To ensure the robustness of the generated code, as mentioned in line 226 of the text: "We filter out new events that occur less frequently than a hyperparameter K and are not in the predefined event category." This is done to ensure that the event knowledge extracted from LLMs is grounded in common sense and remains stable.
>
> Pertaining to the ablation experiments conducted within the IGE module, as illustrated in Figure 7 within the text, our approach in the IGE module involves a sequential integration of event knowledge expansion, temporal relation expansion, and entity relation expansion. Each subsequent module builds upon the foundation of the preceding one, culminating in the acquisition of a comprehensive and usable event knowledge base. Every module is indispensable, rendering it unfeasible to conduct ablation experiments.
>
> **Response to Rejection 2: Authors' experiments were conducted on just one specific event scenario.**
>
> Thanks for your suggestion. Following previous studies, we utilize the same dataset and evaluation metrics for fair comparison. Since no other datasets related to this task is available, we cannot explore model assessment in another scenarios. We acknowledge the importance of comprehensively evaluating our model across different datasets and plan to do so in the future.
>
> **Response to Question 1: Have you designed any experiment to inspect and validate auto-generated codes from LLM?**
>
> Please see the response to Rejection 1.

---

### Official Review · Reviewer_XvjL · 2023-08-07

**Soundness:** 3

**Excitement:**

3: Ambivalent: It has merits (e.g., it reports state-of-the-art results, the idea is nice), but there are key weaknesses (e.g., it describes incremental work), and it can significantly benefit from another round of revision. However, I won't object to accepting it if my co-reviewers champion it.

**Paper Topic And Main Contributions:**

To address the complex event schema induction task, this paper leverages the LLMs to inject knowledge into instance graphs to improve the knowledge coverage of instance graphs, and utilizes discrete diffusion model to tackle error propagation of auto-regressive decoding. Experimental result indicates the proposed KDM outperforms two baselines.

**Questions For The Authors:**

Question A: For Reject 1, why choose code style to build prompt?
Question B: What is the experimental result of Connection Match evaluation in table 1?
Question C: In the experiments, why was there no direct comparison with the predicted results of LLM?
Question D: Has there been any filtering for biases and sensitive information that may arise from large models?
Question E: In the diffusion model, how do you choose the initial random seed, which is very important for the results?
Question F: The experimental results of the generative methods are usually very variable. Is there any exploration of the robustness of the model?


**Reasons To Accept:**

- This paper utilizes the LLMs to enhance the knowledge coverage of instance graphs, which is simple and effective.
- With the help of the ability of large models and generative methods, this article seems to make good use of the advantages of both.
- This paper is well structured, but some details are still worth improving.


**Reasons To Reject:**

- This paper proposes some effective techniques but not provides why choose them, such as IGE module use a Python style object-oriented prompt to extract knowledge from LLMs, but this is not the only way, obviously. So why the authors choses this way needed to be explained or be proved it experimentally?
-  The experimental setting lacks some important baselines, such as LLM direct prediction results; The ablation experiment is insufficient, and the important submodules in the IGE and ESG modules are not explored, such as: event knowledge expansion, temporal relation expansion, and entity relation expansion, et al.
- The details of the experiment are not detailed enough, some hyperparameters cannot find the corresponding settings, and the reproducibility is weak.


**Reproducibility:**

3: Could reproduce the results with some difficulty. The settings of parameters are underspecified or subjectively determined; the training/evaluation data are not widely available.

**Reviewer Confidence:**

4: Quite sure. I tried to check the important points carefully. It's unlikely, though conceivable, that I missed something that should affect my ratings.

**Typos Grammar Style And Presentation Improvements:**

-	In section 4.4, the data lists 4 evaluation metrics, but only discusses sequence event match and connection match evaluation. Entity type match and KL-div of nodes and relations should also be discussed appropriately.
-	Line 492: In Table 3, comparing our KDM model with a variant that removes … There are two variants in the table, which do not seem to correspond.
-	Line 502: There is a grammar error.
-	In Section 4.6, Entity Predictor Ablation Experiment doesn't seem to be very important to explain the contribution of the paper.

---

> ### Author Rebuttal · Authors · 2023-08-28
>
> Thanks for your guidance and suggestions. We will use these as a foundation to further enhance our work. Below is a point-by-point response to your feedback.
>
> **Response to Rejection 1: Why did the paper choose the Python-style object-oriented prompt for the IGE module without providing reasoning or experimental validation?**
>
> Apologize for not explaining the reasons and experiments behind our use of the Python style object-oriented prompt in the IGE module. Based on your advice, we've added the following explanations:
>
> Complex event schemas involve intricate graph structures, while LLMs are good at processing unstructured language tasks. To retain structured information of instance graphs, we need LLMs to be able to handle structured inputs and outputs. Considering the powerful coding capabilities of LLMs, we treat events as Python objects. In detail, events, entities, and their intricate relations can correspond to classes, attributes, and instances in the object-oriented paradigm, respectively. To prove the effectiveness of our approach, we conduct experiments as shown in Table.
>
>
>
> | IGE module             | Event type match(F1) | Event sequence match(l=2) | Event sequence match(l=3) | KL divergence(Node type) | KL divergence(Edge type) | EN    |
> | :--------------------- | -------------------- | ------------------------- | ------------------------- | ------------------------ | ------------------------ | ----- |
> | without IGE            | 0.672                | 0.411                     | 0.216                     | 1.96                     | 3.67                     | -     |
> | IGE with JSON prompt   | 0.685                | 0.418                     | 0.221                     | 1.94                     | 3.89                     | 2     |
> | IGE with Python prompt | **0.713**            | **0.462**                 | **0.268**                 | **1.91**                 | **3.45**                 | **9** |
>
>
>
>
> The table express the results of different prompts for the IGE Module on Suicide-IED dataset. **EN** is the number of effective events generated by the LLM after filtering, which are used for instance graph expansion.
>
> JSON is a prevalent format for representing structured data. We encode the data in JSON format and instruct the LLMs to perform expansion, while maintaining the rest of the process consistent with the Python prompt approach. As shown in the Table, the results obtained through the use of Python prompts are noticeably better than those achieved with JSON prompts. And after filtering, the event types generated by the Python prompt are significantly more numerous than those generated by the JSON prompt. This observation underscores the effectiveness of the Python prompt approach.
>
> **Response to Rejection 2: What are the missing baselines like LLM direct predictions?  The ablation experiment about IGE and ESG modules is insufficient**
>
> 1. Given the unsatisfactory outcomes of using LLMs directly for the task, we opted not to establish it as our baseline. As elaborated in the previous response, we recognize that LLMs struggle with processing structured graph data. To facilitate a more balanced comparison, we add the LLMs' results, as presented in the table below.
>
>    | Dataset     | Model     | Event type match(F1) | Event sequence match(l=2) | Event sequence match(l=3) | KL divergence(Edge type) | KL divergence(Edge type) | CM        |
>    | ----------- | --------- | -------------------- | ------------------------- | ------------------------- | ------------------------ | ------------------------ | --------- |
>    | General-IED | FBS       | 0.614                | 0.199                     | 0.064                     | 2.98                     | 6.13                     | -         |
>    | General-IED | DoubleGAE | 0.627                | 0.266                     | 0.093                     | 2.72                     | 5.84                     | 0.046     |
>    | General-IED | LLM       | 0.520                | 0.176                     | 0.041                     | 2.72                     | 5.84                     | -         |
>    | General-IED | KDM(ours) | **0.704(0.004)**     | **0.380(0.002)**          | **0.181(0.003)**          | **2.32(0.000)**          | **4.53(0.007)**          | **0.185** |
>    | Car-IED     | FBS       | 0.650                | 0.198                     | 0.065                     | 1.86                     | 5.85                     | -         |
>    | Car-IED     | DoubleGAE | 0.654                | 0.285                     | 0.107                     | 2.70                     | 5.71                     | 0.044     |
>    | Car-IED     | LLM       | 0.515                | 0.150                     | 0.031                     | 2.70                     | 6.34                     | -         |
>    | Car-IED     | KDM(ours) | **0.701(0.000)**     | **0.395(0.001)**          | **0.297(0.000)**          | **1.91(0.000)**          | **4.16(0.002)**          | **0.176** |
>    | General-IED | FBS       | 0.626                | 0.210                     | 0.061                     | 2.11                     | 5.59                     | -         |
>    | General-IED | DoubleGAE | 0.624                | 0.210                     | 0.096                     | 2.19                     | 5.33                     | 0.046     |
>    | General-IED | LLM       | 0.493                | 0.174                     | 0.053                     | 2.75                     | 5.65                     | -         |
>    | General-IED | KDM(ours) | **0.713(0.000)**     | **0.462(0.001)**          | **0.268(0.000)**          | **1.91(0.000)**          | **3.45(0.000)**          | **0.176** |
>
>
> In the table, schema matching score (\%) is calculated by checking the intersection of the induced schemas and the manually checked test schemas. The variance of our experiments on three different random number seed is in the brackets.
>
>
> 2.	As illustrated in Figure 7, the IGE module involves a sequential integration of event knowledge expansion, temporal relation expansion, and entity relation expansion. Each subsequent module builds upon the foundation of the preceding one, culminating in the acquisition of a comprehensive and usable event knowledge base. Thus every module is indispensable, rendering it unfeasible to conduct ablation experiments.
>
> **Response to Rejection 3: provide more detailed hyperparameters and improve reproducibility**
>
> Actually, we provide an in-depth exposition of all our implementation details and hyperparameters in Appendix B (i.e., between Line 828 and Line 857). Simultaneously, we are committed to open-sourcing our code for reproducibility and further research.
>
> **Response to Question A: why choose code style to build prompt?**
>
> Please see the response to Rejection 1.
>
> **Response to Question B: What is the experimental result of Connection Match evaluation in table 1?**
>
> Actually, we have elucidated the definition of "CM" between Line 438 and Line 444.
>
> **Response to Question C:  In the experiments, why was there no direct comparison with the predicted results of LLM?**
>
> Please see the response to Rejection 2.
>
> **Response to Question D:  Has there been any filtering for biases and sensitive information that may arise from large models?**
>
> In our experiments, we do not encounter any biases or sensitive information. Therefore, we do not apply filtering process.
>
> **Response to Question E: In the diffusion model, how do you choose the initial random seed, which is very important for the results?**
>
> In our experiments, we utilize three distinct sets of random seeds, all of which ultimately yield converged diffusion models. To underscore the stability of the model, we supplement the experimental variance, as depicted in the Table above. From the result, we can observe that the impact of random seed on the results is relatively minor.
>
> **Response to Question F: The experimental results of the generative methods are usually very variable. Is there any exploration of the robustness of the model?**
>
> Please see the response to Question E.
>
> **Response to Typos Grammar Style And Presentation Improvements**
>
> Thank you very much for pointing out the spelling, grammar, and expression issues. We will correct these errors in the revised version.

---

### Meta-Review · Area_Chair_Fzdg · 2023-09-15

**Recommendation:** 4

**Metareview:**

This paper introduces a knowledge-enriched discrete diffusion model for event schema induction. Further, the authors use an OpenAI LLM  to generate training data. The technologies used here are combined in an interesting and novel way, and the paper robustly demonstrates that the strategy leads to improved performance on several benchmarks.

The model and results presented are interesting. As identified by multiple reviewers, the main flaw of this paper is clarity – the original version left out many details, and important sections were underspecified. This has been partially addressed by authors in responses – i.e., the authors have added motivation for their choice to prompt through code generation. When the paper is updated to reflect the rebuttals, it will be in a much stronger state.

---

### Decision · Program_Chairs · 2023-10-07

**Decision:**

Accept-Findings

**Comment:**

This paper introduces a knowledge-enriched discrete diffusion model for event schema induction. Further, the authors use an OpenAI LLM  to generate training data. The technologies used here are combined in an interesting and novel way, and the paper robustly demonstrates that the strategy leads to improved performance on several benchmarks.

The model and results presented are interesting. As identified by multiple reviewers, the main flaw of this paper is clarity – the original version left out many details, and important sections were underspecified. This has been partially addressed by authors in responses – i.e., the authors have added motivation for their choice to prompt through code generation. When the paper is updated to reflect the rebuttals, it will be in a much stronger state.